# A Composite Indicator for Primary Diabetes Care: A Cross-Sectional Study in Hungary

**DOI:** 10.3390/healthcare13050480

**Published:** 2025-02-22

**Authors:** Undraa Jargalsaikhan, Feras Kasabji, Ferenc Vincze, Anita Pálinkás, László Kőrösi, János Sándor

**Affiliations:** 1Department of Public Health and Epidemiology, Faculty of Medicine, University of Debrecen, H-4012 Debrecen, Hungary; jargalsaikhan.undraa@med.unideb.hu (U.J.); kasabji.feras@med.unideb.hu (F.K.); palinkas.anita@med.unideb.hu (A.P.); 2Doctoral School of Health Sciences, University of Debrecen, H-4012 Debrecen, Hungary; 3Department of Financing, National Health Insurance Fund, H-1139 Budapest, Hungary; korosi.l@neak.gov.hu; 4HUN-REN-DE Public Health Research Group, Department of Public Health and Epidemiology, Faculty of Medicine, University of Debrecen, H-4012 Debrecen, Hungary

**Keywords:** primary care, diabetes mellitus, composite indicator, performance indicators, general medical practice characteristics

## Abstract

**Background:** Since the effectiveness of primary DM care (PDMC) is influenced by both health-care-related and external factors, its indicator set must include indicators that are easy-to-understand for all stakeholders, such as composite indicator-based ranking. **Objectives:** Our study aimed to prepare a composite PDMC indicator, which is adjusted with the GP-independent characteristics of a GMP, in order to evaluate the usefulness of composite indicators in performance-improving programs. **Methods:** Using indicators standardized by age, sex, and eligibility for exemption certificates (on hemoglobin A1C, lipid status, serum creatinine, and ophthalmological examination, and on influenza vaccination) for Hungarian adult DM care, factor analysis was applied to create a composite DM care quality indicator (CDMI). It was adjusted (ACDMI) by a multivariable linear regression model of the association between structural characteristics of GMPs and the CDMI. **Results:** There were 516,052 DM patients provided for by 4784 GMPs. The CDMI exhibited significant associations with patients’ lower education (β = −0.139, 95%CI: −0.182; −0.095), GPs’ age over 65 (β = −0.083, 95%CI: −0.109; −0.056), GMPs with more than 2000 adult patients (β = −0.059, 95%CI: −0.090; −0.027), and urban location (β = 0.096, 95%CI: 0.058; 0.134). The average difference in GMPs’ ranks by the CDMI and ACDMI was 583. Extreme poor (N = 147) and extreme good (N = 176) performances of GMPs were identified, and those were categorized further by the role of GP-independent factors in causing the extreme performances (N_healthcareunrelated_ = 84; N_healthcarerelated_ = 239). **Conclusions:** Our findings suggest a stepwise and widely communicable process for PDMC monitoring, which starts with the evaluation of the CDMI and ACDMI to identify the GMPs requiring interventions, making a distinction between extreme GMPs requiring health-care-related interventions and those requiring non-health-care-related interventions.

## 1. Introduction

Diabetes mellitus (DM) is one of the most common causes of health issues. It generates enormous health care costs worldwide. The global prevalence of DM has increased to more than 422 million affected individuals and is expected to increase further in the next two decades, reflecting limitations in preventive efforts in practice [1,2].

As the scientific base of DM care grows and the guidelines provide increasingly effective treatment, adherence to guidelines becomes increasingly important. Therefore, health care providers are searching for measures that make DM care transparent and improve its quality [3].

There conceivably is an indicator set that is informative for experts, who can decipher the intricacies of DM care and can specify the necessary interventions. However, experience indicates that the information provided by an indicator set cannot be handled by patients and decision makers, who are to identify only the need for intervention (but not the need for a specific intervention in a certain part of DM care). For these nonexperts, the indicator set must be simplified and translated into a single evaluative score [4,5,6].

Composite indicators summarize information on different elements of care to comprehensively measure DM care quality. Comparative evaluations, ranking protocols, performance tracking, benchmarking, pay-for-performance (P4P) systems, and public reporting require these types of indicators instead of the complex review of a set of indicators focused on the details of DM care [7,8,9,10]. However, developing composite indicators faces challenges ranging from selecting component indicators, through weighting and aggregation, to ensuring the overall validity of the composite score produced [7].

In Hungary, a P4P model was introduced into primary care in 2009 by the National Health Insurance Fund (NHIF). It is the only health insurance fund that is contracted with each general medical practice (GMP) in the country. The objective of this initiative was to motivate general practitioners (GPs) to deliver high-quality care with a focus on particular chronic conditions, including DM [11,12,13]. Two elements of the implemented indicator set focused on DM care quality: (1) the percentage of individuals with DM in the GMP who had undergone hemoglobin A1c (HbA1c) testing at least once within the past 12 months and (2) the percentage of individuals with DM in the GMP who had undergone ophthalmological examination at least once within the past 12 months. This monitoring system has been operating continuously. Monthly feedback is provided to GMPs by the NHIF.

However, despite these efforts, the performance of primary care and the quality of DM care in Hungary are still relatively poor compared with other European countries [14,15,16]. The necessary intervention is probably multimodal and also requires the strengthening of monitoring methodology. The NHIF has an extensive database that could be leveraged to create more specific indicators, which could be converted into a composite DM care quality indicator (CDMI), supporting both experts and nonexperts [17,18,19,20].

The aims of this study were (1) to extend the NHIF’s set of DM-focused indicators and to create a CDMI for GMPs across Hungary, focusing on adult patients; (2) to examine the relationships between the CDMI and the structural characteristics of GMPs in order to create the adjusted CDMI, which is corrected for the impact of the GP-independent structural characteristics of GMPs; and (3) to offer recommendations for improving monitoring and monitoring based organized interventions.

## 2. Materials and Methods

### 2.1. Study Settings

Our cross-sectional retrospective analysis included all the GMPs in Hungary, which totaled 4784 GMPs. This was a secondary database analysis with a study period of April 2017–March 2018. Patients with DM were identified via the NHIF’s criteria, which require a minimum of four DM medicine redemptions (ATC code: A10) within one year. Data on DM patient care were provided by the NHIF by sex and age groups (18–19, 20–24, 25–29, 30–34, 35–39, 40–44, 45–49, 50–54, 55–59, 60–64, 70–74, 75–79, 80–84, 85–89, and 90≤) for each GMP. Data regarding GMP structural parameters were extracted from the NHIF’s integrated information system. Socioeconomic status (SES) data were sourced from the 2011 Hungarian Census, which was provided by the Hungarian Central Statistical Office (HCSO).

The study adhered to Strobe Statement guidelines for reporting cross-sectional studies [21].

### 2.2. Explanatory Variables

The structural characteristics of the GMPs were delineated by the number of patients under GMP care (categorized as ≤800, 801–1200, 1201–1600, 1601–2000, and ≥2001), GP status (distinguishing between GMPs managed by temporary GPs with limited availability and those with contracted GPs in permanent positions; GPs were classified further by age as <65 or ≥65 years), type of settlement (urban or rural), and geographical location by county. The prevalence of DM in a GMP was computed for persons aged 40–54 years and for persons aged 55–69 years. Settlement-level SES indicators were derived from census data via indicators such as Carstair’s score components [22,23,24], which were computed as follows:Indirectly standardized relative level of education (srEDU) and standardized relative employment ratio (srEMP), as summarized by years of school attendance and the number of employed persons over the expected values on the basis of age- and sex-specific national references and demographic data of the settlement.Relative housing density (rHD) and relative Roma proportion (rRP) are the number of occupants per room over the average Hungarian housing density and the proportion of self-declared Roma individuals in a settlement divided by the population share of self-declared Roma individuals in Hungary, respectively.

GMPs may provide care for adults residing in various areas. GMP-specific SES indicators were determined via the weighted average of settlement-specific SES indicators. Weights were obtained from the distribution of adult residential places belonging to a GMP. The GMP-level SES indicators were sorted into tertiles.

### 2.3. Composite Outcome Variable

We created a CDMI using 5 process indicators. These indicators included the proportion of DM patients who underwent HbA1C testing, ophthalmological examination, serum creatinine measurement, serum lipid status measurement, and the proportion of DM patients under 65 years of age who had received an influenza vaccination within the past 12 months [25,26,27,28,29]. The GMP-specific DM care indicators were indirectly standardized, taking into account age categories, sex, and eligibility for exemption certificates. (These certificates, granted by local municipalities, enable free access to medications for people with low socioeconomic status and chronic conditions.) The expected GMP-specific numbers were determined via national stratum-specific reference values and the demographic composition of the population provided by each GMP, resulting in indirect standardized ratios (ISRs) computed for each indicator and GMP.

To address the issue of low observed case numbers in the statistical analysis, the ISRs were transformed via empirical Bayes adjustment. The empirical Bayes-adjusted ISRs were then normalized via the two-step Box–Cox method (nISRs). The nISRs were subjected to factor analysis via principal component analysis with PROMAX rotation. The number of factors was determined by the eigenvalues. An eigenvalue greater than 1 was the criterion. The analysis leads to the 1-factor outcome (Kaiser–Meyer–Olkin measure = 0.727; Bartlett’s test *p* < 0.001). (Factor loadings are shown in Appendix A Table A1). The standardized outcome of this factor analysis (composite DM indicator and CDMI) was used to describe the quality of DM care in the GMPs.

### 2.4. Statistical Analysis

Multivariable linear regression models were used to investigate the relationships between the GMP features and the CDMI. To demonstrate the relative impact of the explanatory variables, standardized linear regression coefficients (β) and their corresponding 95% confidence intervals (95% CI) were computed.

Using the nonstandardized linear regression coefficients from the regression model, the effects of the explanatory variables were removed from the CDMI values. The distribution of the adjusted CDMI values (ACDMI) was compared with the distribution of the CDMI values. GMPs with extreme positions were identified as those with scores falling below the 2.5th percentile (extremely low) or falling above the 97.5th percentile (extremely high). The extreme GMPs were categorized as follows:GMPs with extremely low CDMI and extremely low ACDMI, where the low score was not attributable to the structural factors investigated in the regression model (low DM care quality elicited by nonstructural factors: lowCDMI_nonstructural_);GMPs with extremely low CDMI but not extremely low ACDMI, where the low score was attributable to the structural factors investigated in the regression model (low DM care quality elicited by structural factors: lowCDMI_structural_);GMPs with extremely high CDMI and extremely high ACDMI, where the high score was not attributable to the structural factors investigated in the regression model (high DM care quality elicited by nonstructural factors: highCDMI_nonstructural_);GMPs with extremely high CDMI and not extremely high ACDMI, where the high score was attributable to the structural factors investigated in the regression model (high DM care quality elicited by structural factors: highCDMI_structural_).GMPs with not extreme CDMI but extremely high ACDMI, where the high adjusted score was not attributable to the structural factors investigated in the regression model (high DM care quality elicited by nonstructural factors: highACDMI_nonstructural_).GMPs with not extreme CDMI but extremely low ACDMI, where the low adjusted score was not attributable to the structural factors investigated in the regression model (low DM care quality elicited by nonstructural factors: highACDMI_nonstructural_).

PASW Statistics (version 18.0, SPSS Inc., Chicago, IL, USA) was utilized for data analysis.

## 3. Results

### 3.1. Sample

We identified 516,052 DM patients with a mean age of 65.73 years (SD = 12.37 years) and a 1.11 male/female ratio. A total of 8.34% of the DM patients had an exemption certificate.

The majority of both GMPs (66.30%) and DM patients (70.80%) were in urban areas. Most GMPs (63%) had a list size between 1201 and 2000 patients, and DM patients were also concentrated in GMPs with similar list sizes (62.22%). A small proportion of GMPs had GP vacancies (3.70%), accounting for 2.40% of DM patients. Overall, 22.11% of the GPs were at least age 65 years and were responsible for 20.12% of the DM patients (Table 1).

### 3.2. Utilization of DM Care Services

The proportions of DM patients who properly participated in HbA1C testing, ophthalmologic examination, serum creatinine testing, and lipid status checking and who received influenza vaccinations were 78.05% (402,783/516,052), 38.03% (196,278/516,052), 86.18% (444,715/516,052), 78.05% (402,784/516,052), and 12.89% (28,131/218,264), respectively. Overall, 58.64% (302,613/516,052) of the DM patients properly received the investigated five services. The distribution of the composite indicator (mean = 2.91 × 10^−17^, SD = 0.996; N = 4784) is shown in Figure 1.

### 3.3. Linear Regression Modeling

The regression model (r^2^ = 0.145; the distribution of the standardized residuals is shown in Appendix A Figure A1) demonstrated that both patient and GMP characteristics had significant impacts on the CDMI (Table 2).

A low level of education emerged as the most influential determinant for the CDMI (β = −0.139, 95% CI: −0.182 to −0.095). In addition, a GP over age 65 years (β = −0.083, 95% CI = −0.109 to −0.056) and a large GMP size (β = −0.059, 95% CI: −0.090 to −0.027) were also significant negative factors impacting the CDMI. Residing in an urban area (β = 0.096, 95% CI: 0.058 to 0.134) and having a high level of education (β = 0.057, 95% CI: 0.011 to 0.102) had positive effects on the CDMI. The geographical inequality described by the regression models was large for the CDMI (Appendix A Table A2). The prevalence of DM, Roma ethnicity, housing density, and employment were not associated with the CDMI.

The distributions of the CDMI and ACDMI were similar (Figure 1), but the changes in GMP ranks were remarkable. (Figure 2) The average rank change (the SD for rank change distribution) was 583.

There were 91 and 28 GMPs where the extremely low CDMI could be attributed to nonstructural (lowCDMI_nonstructural_) and structural factors (lowCDMI_structural_), respectively. Extremely high CDMIs for 120 GMPS were attributable to structural factors in the case of 56 GMPs (highCDMI_structural_) and to nonstructural factors for 64 GMPs (highCDMI_nonstructural_). There were 28 GMPs with a not-extreme CDMI, but an extremely low structural-factors-adjusted performance (highACDMI_nonstructural_). The adjusted performance was extremely high for another 56 GMPs with a not-extreme CDMI (highACDMI_nonstructural_). (Figure 3; a more detailed description of the DM care performance in the groups of GMPS is summarized in Appendix A, Table A3).

## 4. Discussion

### 4.1. Main Findings

Our investigation demonstrated that it is technically possible to develop a CDMI using NHIF data in Hungary, avoiding the necessity for primary data collection. The composite indicator we developed explains 59.1% of the total variance of DM care across the practices’ quality covered by the five distinct indicators (the CDMI for nonexpert-level decisions holds considerable, but not all, information on the specific indicators). The CDMI-based ranking seems to be meaningful and useful. Since diabetes primary care is organized by well-established international recommendations, similar composite DM indicators can be constructed via established frameworks from reputable sources, such as the DIABCARE Q-Net, the NHS and QOF, the EUBIROD guidelines, the National Diabetes Services Scheme, the Healthcare Effectiveness Data and Information Set, and the Robert Koch Institute’s indicator system [13,18,30,31,32].

We also identified several factors that cannot be influenced by GPs but are associated with the CDMI; these were identified via regression modeling. Among these factors, the strongest impact was the low level of patients’ education, which is strongly interrelated with many well-known risk factors for poor DM outcomes (e.g., low health literacy, low income, and improperly understood disease management strategies [33,34]). Another significant factor was the age of the GPs. Those over 65 years of age were associated with lower CDMI scores. This might be because older GPs struggle to keep pace with the latest DM management protocols, management strategies, or treatment strategies [35,36,37]. A larger size of the GMP was shown to also be a risk factor. The larger GMPs achieved a reduced CDMI score. The explanation is, at first glance, simple overloading of the capacities of the GMP [38,39]. Rural residential places were found to contribute negatively to CDMI. Urban populations typically have better access to health care, higher levels of health awareness, and complementary resources that promote good DM care [40,41,42]. Finally, our study confirmed the usual observations from other countries that there are significant regional differences in DM care quality [42,43,44].

Some determinants, including DM prevalence, Roma ethnicity, housing density, and employment status, did not reveal a notable association with the CDMI. These factors likely do not play important roles in the quality of DM care, or their effects could be offset by other parameters, such as education or the availability of services.

Our study also highlighted the importance of adjusting the CDMI and distinguishing among GMPs with extreme ranks according to the role of GP-dependent and GP-independent DM care influencing factors. This approach can help to provide a more meaningful picture of GP performance based on factors within their control. When compared to others, the ranking differed substantially when we controlled for the age of the GP or rural status of the practice, for example. This suggests that P4P must account for these external factors to ensure a fair motivation of GPs and avoid punishing them for unmodifiable circumstances. With this approach, the P4P system, which can impact the GP’s personal contribution, could be targeted more effectively.

### 4.2. Strengths and Limitations

This study’s strength was the inclusion of all Hungarian GMPs and the entire adult population of Hungary, which prevented selection bias and ensured high statistical power. In addition, standardized data collection methods employed by the NHIF and the HCSO minimized the risk of misclassification of the studied parameters.

This study has several limitations. First, many aspects of DM care and the type of DM were not taken into consideration in computing the CDM because the availability of input data was restricted. Second, the cross-sectional design restricts the ability to draw conclusions on causal links between explanatory variables and the CDMI. This restricts the reliability of the adjustment based on the linear regression model. Third, unmeasured confounding factors that influence the effectiveness of GPs’ work, such as patients’ health literacy, lifestyle, attitudes [45,46,47,48,49], and comorbidities [50], further weakened the completeness of the regression model and the adjustment. Consequently, the role of GPs’ personal contribution was overestimated, whereas the role of external factors was underestimated in our analysis.

Furthermore, the SES indicators were generated from 2011 data, whereas the GMP structural parameters and DM indicators were derived from 2018 data. Because socioeconomic changes occur gradually, this 7-year gap may have had a slight influence on the observed results, probably resulting in weaker observed associations.

### 4.3. Perspectives for Clinical Practice

The ultimate purpose of monitoring GMPs is to enhance patient outcomes. Within the framework of the monitoring, identifying the GMPs that need intervention and specifying the required intervention are different, consecutive tasks. Our study demonstrated that it is possible to prepare easy-to-understand composite indicators for DM care by processing the existing health records of DM patients, and these indicators can be utilized in a stepwise approach.

In the primary evaluation, the GMPs can be ranked by the CDMI and the extreme GMPs can be identified (where patients are provided with better or weaker care); next, GMPs can be also ranked according to the CDMI adjusted for GP independent factors. Considering both rankings together, GMPs with an extreme performance can be identified and we can make distinction between those GMPs where the extreme performance was elicited by extreme environment and those GMPs where the extreme performance originated from the extreme performance of the GMP staff or from external factors not involved in the regression model. This primary evaluation can be managed by the NHIF. Reporting these results could improve the transparency of DM care through measures that are understandable even to common people. This transparency could motivate both GPs and other stakeholders to undertake interventions by which DM care could be improved.

In the secondary evaluation, a GP alone or with other stakeholders can draw practical conclusions. If the GP-independent factors were responsible for poor GMP performance then the evaluation process has to involve the external stakeholders. In this setting, the improvement of the working environment of the GMPs should be targeted by the discussion. If the GMP staff performance established the extreme CDMI or ACDMI, then the elimination of the malpractice or benchmarking has to be encouraged. It can be supported by P4P schemes based on the adjusted CDMI. (According to our analysis, adjusted and nonadjusted ranks of GMPs differ considerably. This emphasizes the importance of the use of adjusted indicators in P4P systems to avoid mistargeted financial interventions.) Depending on the local setting, diabetes case manager nurses [51,52] or public health specialists [53,54,55] or other health professionals supplementing the traditional primary care team could play a meaningful role in the secondary evaluation if their list of competences was supplemented by the local interpretation of the report on the performance indicators and the preparation of the local intervention plan.

## 5. Conclusions

A composite measure for DM primary care can be computed utilizing the existing NHIF data in Hungary, which can be used to rank GMPs and to identify the GMPs where intervention is needed or benchmarking is possible. According to the composite indicator, DM care is less effective in rural areas, in larger practices managed by older GPs, and in GMPs providing care for less educated adults. These GP-independent factors considerably affect the composite indicator; therefore, P4P schemes should use the adjusted version of the composite indicators to properly target the motivation of GPs. Our results suggested that a stepwise process of performance evaluation for DM primary care could start with composite and adjusted composite measure evaluations to identify the GMPs requiring interventions in a manner that is transparent to nonexperts as well. This could strengthen the motivation for both GPs and other stakeholders to participate in a thorough evaluation and in the implementation of expert recommendations.

## Figures and Tables

**Figure 1 healthcare-13-00480-f001:**
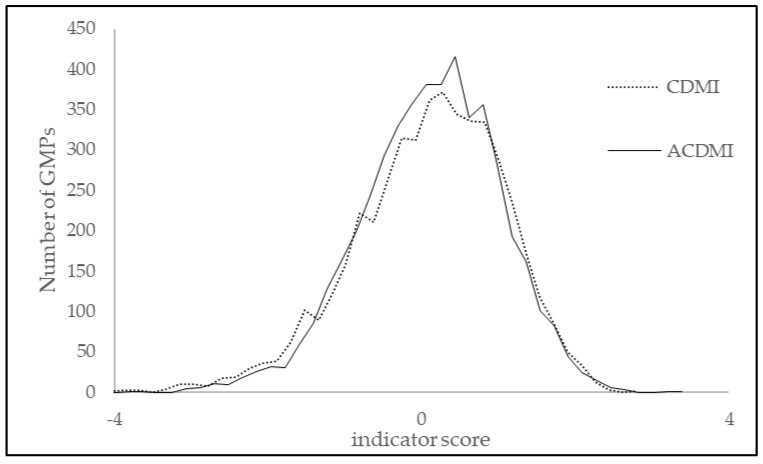
Distribution of the composite DM care quality indicator (CDMI) and the adjusted composite DM care quality indicator (ACDMI) for Hungarian GMPs.

**Figure 2 healthcare-13-00480-f002:**
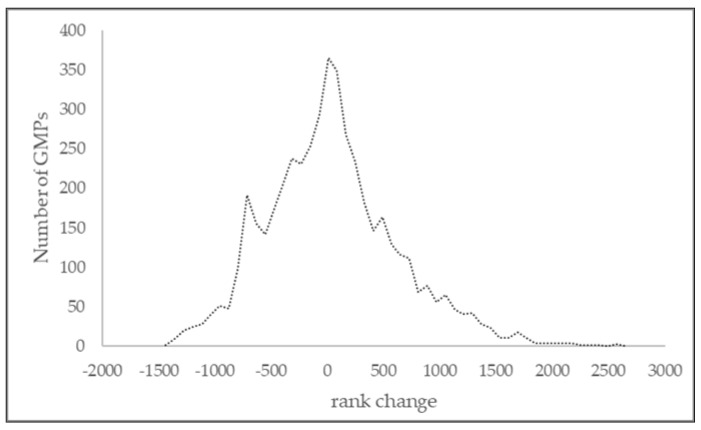
Distribution of the difference between general medical practices (GMPs) position (rank_ACDMI_—rank_CDMI_) in the rank for the composite DM care quality indicator (CDM) and for the adjusted composite DM care quality indicator (ACDMI).

**Figure 3 healthcare-13-00480-f003:**
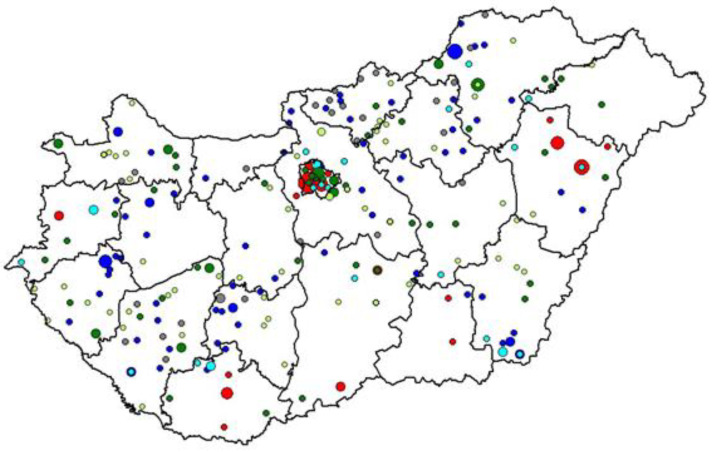
Spatial distribution of general medical practices (GMPs) with extreme composite indicators by the explanatory role of the structural characteristics of GMPs. (Dark blue: GMPs with extremely low CDMI and extremely low ACDMI, lowCDMI_nonstructural_. Grey: GMPs with extremely low CDMI but not extremely low ACDMI, lowCDMI_structural_. Dark green: GMPs with extremely high CDMI and extremely high ACDMI, highCDMI_nonstructural_. Red: GMPs with extremely high CDMI and not extremely high ACDMI, highCDMI_structural_. Light green: GMPs with not extreme CDMI but extremely high ACDMI, highACDMI_nonstructural_. Light blue: GMPs with not extreme CDMI but extremely low ACDMI, lowACDMI_nonstructural_. Size of the symbol is proportional to the number og GMPs with certain characteristics in a settlement).

**Table 1 healthcare-13-00480-t001:** Distribution of general medical practices (GMPs) and DM patients by structural characteristics of the investigated GMPs.

Structural Parameters	Categories	GMP	DM Patients
Types of settlement	Urban	3172 (66.30%)	365,381 (70.80%)
	Rural	1612 (33.70%)	150,671 (29.20%)
List sizes of GMP	<800	153 (3.00%)	7207 (1.40%)
	801–1200	655 (14.00%)	45,335 (8.78%)
	1201–1600	1522 (32.00%)	142,544 (27.62%)
	1601–2000	1504 (31.00%)	178,537 (34.60%)
	>2000	950 (20.00%)	142,429 (27.60%)
Vacancy of GP	Filled	4608 (96.30%)	503,713 (97.60%)
	Vacant	176 (3.70%)	12,339 (2.40%)
Age of GP (years)	≥65	1019 (22.11%)	103,895 (20.13%)
	<65	3589 (77.89%)	399,818 (77.47%)
Total	4784 (100%)	516,052 (100%)	

**Table 2 healthcare-13-00480-t002:** Influence of general medical practices’ (GMPs) structural characteristics and patients’ socioeconomic status on the standardized composite DM care quality indicator of GMPs and their corresponding 95% confidence intervals according to multivariable linear regression analysis.

Explanatory Variables	Types	β (95%CI *)
Type of settlement	Urban	**0.096 [0.058; 0.134]**
	Rural	Reference
GP	Vacancy	0.011 [−0.018; 0.041]
	age ≥ 65	**−0.083 [−0.109; −0.056]**
	age < 65	Reference
List size of GMP	≤800	−0.025 [−0.054; 0.004]
	801–1200	−0.026 [−0.055; 0.003]
	1201–1600	Reference
	1601–2000	−0.013 [−0.044; 0.018]
	2001<	**−0.059 [−0.090; −0.027]**
Level of patients’ education	Low	**−0.139 [−0.182; −0.095]**
	Medium	Reference
	High	**0.057 [0.011; 0.102]**
Employment ratio among patients	Low	−0.020 [−0.066; 0.025]
	Medium	Reference
	High	0.011 [−0.039; 0.061]
Housing density	Low	0.016 [−0.025; 0.057]
	Medium	Reference
	High	−0.022 [−0.059; 0.016]
Roma proportion	Low	0.004 [−0.041; 0.049]
	Medium	Reference
	High	0.002 [−0.047; 0.050]
Prevalence of DM	among 40–54-year-olds	−0.021 [−0.050; 0.008]
	among 55–69-year-olds	0.021 [−0.009; 0.051]

* Adjusted standardized linear regression coefficients and 95% confidence intervals. The values are adjusted for counties. The whole model is presented in Appendix A Table A2. Significant results in bold.

## Data Availability

The datasets used and/or analyzed during the current study are available from the corresponding author on reasonable request.

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
