# Peer review of "A Composite Indicator for Primary Diabetes Care: A Cross-Sectional Study in Hungary"

_healthcare, 2025, doi:10.3390/healthcare13050480_

Round 1
Reviewer 1 Report (Previous Reviewer 1)
Comments and Suggestions for Authors
1 . What is the main question addressed by the research?
This study tried to present a A composite indicator for primary diabetes care using A cross-sectional study in Hungary. The authors present this indicator like a solution, like a tool for the PDMC monitoring, which starts with the evaluation of CDMI and ACDMI to identify the GMPs requiring interventions, making distinction between extreme GMPs requiring health-care-related and non-healthcare-related interventions.
- Do you consider the topic original or relevant in the field? Does it address a specific gap in the field?
The presented model of indicator is easy to understand and a useful tool for Hungary, and not for them. In the same time, this tool should be used in a longitudinal study to evaluate if it is sensitive to local improvements.
- What does it add to the subject area compared with other published material?
Seems to be a new tool in this area. This is very well presented, very simple structured.
- What specific improvements should the authors consider regarding the methodology? What further controls should be considered?
The methodology is adequate and well expressed. It is easy to understand and—from the previous version—it is easy to catch the study idea.
- Are the conclusions consistent with the evidence and arguments presented and do they address the main question posed?
Yes, the conclusions are consistent, and study could be published in actual shape.
- Are the references appropriate?
References are appropriate. All references are carefully chosen from impact journals. They are updated, from the last years, in the area of public health. This article is a real example for other contributors for HEALTHCARE.
- Please include any additional comments on the tables and figures.
Figures and tables are very easy to understand, presenting the results from HUNGARY. In the same time, the authors could explore other countries to apply their indicator.
Final CONCLUSION—ready to be published.
Author Response
Dear Editor and Reviewer,
Thank you very much for the careful review of our manuscript. Please find enclosed the revised version of the manuscript “A composite indicator for primary diabetes care: A cross-sectional study in Hungary” by Undraa Jargalsaikhan, et al.
Each comment and suggestion has been considered. The corresponding changes and refinements made in the revised paper are summarized in our response after considering each of your suggestion. Answers along with the modifications we made are summarized below (comments/questions are in bold capitals).
Sincerely yours, Janos Sandor (on behalf of the authors)
Answers/reflections to the comments of Reviewer-1:
1.
THIS STUDY TRIED TO PRESENT A COMPOSITE INDICATOR FOR PRIMARY DIABETES CARE USING A CROSS-SECTIONAL STUDY IN HUNGARY. THE AUTHORS PRESENT THIS INDICATOR LIKE A SOLUTION, LIKE A TOOL FOR THE PDMC MONITORING, WHICH STARTS WITH THE EVALUATION OF CDMI AND ACDMI TO IDENTIFY THE GMPS REQUIRING INTERVENTIONS, MAKING DISTINCTION BETWEEN EXTREME GMPS REQUIRING HEALTH-CARE-RELATED AND NON-HEALTHCARE-RELATED INTERVENTIONS.
Thank you for the concise summary of our objectives.
2.
THE PRESENTED MODEL OF INDICATOR IS EASY TO UNDERSTAND AND A USEFUL TOOL FOR HUNGARY, AND NOT FOR THEM. IN THE SAME TIME, THIS TOOL SHOULD BE USED IN A LONGITUDINAL STUDY TO EVALUATE IF IT IS SENSITIVE TO LOCAL IMPROVEMENTS.
This study was a cross-sectional study with acknowledged limitation. We would like to establish a follow-up study where the variability of the composite indicator in time could be investigated. Depending the success of our applications we will carry out this investigation.
3.
SEEMS TO BE A NEW TOOL IN THIS AREA. THIS IS VERY WELL PRESENTED, VERY SIMPLE STRUCTURED.
Thank you for this evaluation.
4.
THE METHODOLOGY IS ADEQUATE AND WELL EXPRESSED. IT IS EASY TO UNDERSTAND AND—FROM THE PREVIOUS VERSION—IT IS EASY TO CATCH THE STUDY IDEA.
Thank you very much for suggestions in the previous round of per-reviewing. We have to admit that the paper improved remarkably. Thanks!
5.
YES, THE CONCLUSIONS ARE CONSISTENT, AND STUDY COULD BE PUBLISHED IN ACTUAL SHAPE.
Thank you for this comment!
6.
REFERENCES ARE APPROPRIATE. ALL REFERENCES ARE CAREFULLY CHOSEN FROM IMPACT JOURNALS. THEY ARE UPDATED, FROM THE LAST YEARS, IN THE AREA OF PUBLIC HEALTH. THIS ARTICLE IS A REAL EXAMPLE FOR OTHER CONTRIBUTORS FOR HEALTHCARE.
Thank you for this comment!
7.
FIGURES AND TABLES ARE VERY EASY TO UNDERSTAND, PRESENTING THE RESULTS FROM HUNGARY. IN THE SAME TIME, THE AUTHORS COULD EXPLORE OTHER COUNTRIES TO APPLY THEIR INDICATOR.
The sentence on the other indicator systems in different countries has been extended, to emphasize more the opportunities for computing primary DM care composite indicators in other countries.
Line 248-253
Original sentence:
“Similar composite DM indicators can be constructed via established frameworks from reputable sources, such as the DIABCARE Q-Net, the NHS and QOF, the EUBIROD guidelines, the National Diabetes Services Scheme, the Healthcare Effectiveness Data and Information Set, and the Robert Koch Institute’s indicator system [13,18,29–32].”
Modified sentence:
“Since, diabetes primary care is organized by well established international recommendations, similar composite DM indicators can be constructed via established frameworks from reputable sources, such as the DIABCARE Q-Net, the NHS and QOF, the EUBIROD guidelines, the National Diabetes Services Scheme, the Healthcare Effectiveness Data and Information Set, and the Robert Koch Institute’s indicator system [13,18,29–31].”

Reviewer 2 Report (Previous Reviewer 2)
Comments and Suggestions for Authors
Dear Authors,
the comments in the annex file.
Best

Author Response
Dear Editor and Reviewer,
Thank you very much for the careful review of our manuscript. Please find enclosed the revised version of the manuscript “A composite indicator for primary diabetes care: A cross-sectional study in Hungary” by Undraa Jargalsaikhan, et al.
Each comment and suggestion has been considered. The corresponding changes and refinements made in the revised paper are summarized in our response after considering each of your suggestion. Answers along with the modifications we made are summarized below (comments/questions are in bold capitals).
Sincerely yours, Janos Sandor (on behalf of the authors)
Answers/reflections to the comments of Reviewer-2:
1.
ABSTACT: I SUGGEST IT TO 250-300 WORDS. THE CONTENT IS GOOD.
We have checked again the number of words in the abstract. That is 250; so there was no correction. Thank you for the general evaluation of the abstract. We have to acknowledge that the former version of the abstract was poor. Thanks for the suggestions to improve that.
2.
KEYWORDS: I SUGGEST REDUCING TO 4-5, EVEN THOUGH THE EDITOR ALLOWS UP TO 8-10.
The redundant mentioning of “monitoring” has been removed from the keywords’ section, according to your suggestion.
3.
METHODS: THIS IS THE MOST CONTROVERSIAL ASPECTS AND CERTAINLY REQUIRES MORE ATTENTION FOR TYPE OF STUDY CONDUCTED, SPECIFICALLY THE LACK OF STRUCTURED REPORTING METHOD, SUCH AS THE STROBE CHECKLIST (DOI:10.1016/J.JCLINEPI.2007.11.008). INCLUDING THIS CHECKLIST AS A SUPPLEMENTARY FILE AND CITING IT IN THE TEXT WOULD BE ESSENTIAL TO IMPROVE THE QUALITY AND REPRODUCIBILITY OF THE STUDY, MAKING THE MANUSCRIPT MORE TRANSPARENT AND SCIENTIFICALLY VALID.
We carried out the investigation and prepared the manuscript considering the STROBE statement for cross-sectional studies. Our manuscript meets each criterion apart from the following:
- Statistical methods; item 12(c); Explain how missing data were addressed: because, there were no missing data in our study
- Statistical methods; item 12(e); Describe any sensitivity analyses: because, we did not apply sensitivity analysis
- Participants, item 13(c); Consider use of a flow diagram: because the sampling was simple (one-stage process) there was no need for a flow chart
- Descriptive data, item 14(b); Indicate number of participants with missing data for each variable of interest: because, there were no missing data in our study
- Main results; item 16(c); If relevant, consider translating estimates of relative risk into absolute risk for a meaningful time period: because it was not relevant for our investigation
- Other analyses;item 17; Report other analyses done—eg analyses of subgroups and interactions, and sensitivity analyses: because there was no analysis of this kind
4.
RESULTS: GOOD, BUT THE LEGENDS IN THE TABLES ARE POORLY PLACED AND NEED RE-EVALUATED BASED ON THE USE OF ACRONYMS.
Thanks for this suggestion! We checked carefully the formatting requirements of the Journal, and corrected the formatting accordingly. Regarding the titles of the tables, the GMP was defined at first mentioning.
Original titles:
Table 1. Distribution of GMPs and DM patients by structural characteristics of the investigated GMPs.
Table 2. Influence of GMPs’ structural characteristics and patients’ socioeconomic status on the standardized composite DM care quality indicator of GMPs and their corresponding 95% confi-dence intervals according to multivariable linear regression analysis.
Figure 2. Distribution of the difference between GMPs’ position (rankACDMI – rankCDMI) in the rank for the composite DM care quality indicator (CDM) and for the adjusted composite DM care quality indicator (ACDMI).
Figure 3. Spatial distribution of GMPs with extreme composite indicators by the explanatory role of the structural characteristics of GMPs.
Modified titles:
Table 1. Distribution of general medical practices (GMPs) and DM patients by structural charac-teristics of the investigated GMPs.
Table 2. Influence of general medical practices’ (GMPs) structural characteristics and patients’ so-cioeconomic status on the standardized composite DM care quality indicator of GMPs and their corresponding 95% confidence intervals according to multivariable linear regression analysis.
Figure 2. Distribution of the difference between general medical practices (GMPs) position (rankACDMI – rankCDMI) in the rank for the composite DM care quality indicator (CDM) and for the adjusted composite DM care quality indicator (ACDMI).
Figure 3. Spatial distribution of general medical practices (GMPs) with extreme composite indicators by the explanatory role of the structural characteristics of GMPs.
5.
DISCUSSION: THE LAST SECTION, WHICH I WOULD HONESTLY CALL “4.3 PERSPECTIVES FOR CLINICAL PRACTICE” COULD BENEFIT FROM EXPANDING THE DISCUSSION ON THE PROPER MULTI-DIMENSIONAL AND MULTIDISCIPLINARY MANAGEMENT OF PATIENTS WITH TYPE 2 DIABETES. IN THIS REGARD, I WOULD LIKE TO SUGGEST TWO HIGHLY RELEVANT THEMES: “LIFESTYLE MEDICINE CASE MANAGER NURSES FOR TYPE TWO DIABETES PATIENTS” AND “NURSE CASE MANAGER LIFESTYLE MEDICINE (NCMLM) IN TYPE TWO DIABETES PATIENT”, WHICH WOULD CERTAINLY EXPAND HE DEBATE ON THE SCIENTIFIC DISCUSSION PROPOSED, WITH STRONG ORGANIZED IMPACT.
Thank you very much for this suggestion! The title of the section has been changed, and the suggested sentence has been added to the section, accordingly.
Original text:
4.3. Implications
Modified text:
4.3. Perspectives for Clinical Practice
[…]
Depending on the local setting, diabetes case manager nurses [50,51] or public health specialists [52-54] or other health professionals supplementing the traditional primary care team could play a meaningful role in the secondary evaluation if their list of competences were supplemented by the local interpretation of the report on the performance indicators and the preparation of the local intervention plan.
6.
LIMITATIONS AND CONCLUSIONS: FINE.
Thanks! The tables have been modified in the appendix as it was done in the main text’s tables.
7.
BIBLIOGRAPHY: IT SHOULD BE EXPANDED ACCORDING TO THE PREVIOUS SUGGESTIONS, AND IT MIGHT BE HELPFUL TO UPDATE REFERENCES OLDER THAN 15-20 YEAR, UNLESS THEY ARE METHODOLOGICAL OR HAVE STRONG IMPACT EVIDENCE.
As we wrote in the former round of peer-reviewing, there were 50 references in the manuscript: 27 published in the last 5 years, 15 published 6 to 10 years before. Regarding the old references:
- Schmittdiel J, Vijan S, Fireman B, et al. Predicted Quality-Adjusted Life Years as a Composite Measure of the Clinical Value of Diabetes Risk Factor Control. Medical Care. 2007;45:315. doi: 10.1097/01.mlr.0000254582.85666.01
- Mannion R, Davies H, Marshall M. Impact of star performance ratings in English acute hospital trusts. J Health Serv Res Policy. 2005;10:18–24. doi: 10.1177/135581960501000106
Our intention by these references (4-5) was to emphasize that the preparation of composite indicators is not new.
- Kringos D, Boerma W, Bourgueil Y, et al. The strength of primary care in Europe: an international comparative study. Br J Gen Pract. 2013;63:e742–50. doi: 10.3399/bjgp13X674422
To our best knowledge, this comprehensive analysis (and project behind) was the last which evaluated and compared the primary care systems in the European countries.
- Ben-Shlomo Y, White I, McKeigue PM. Prediction of general practice workload from census based social deprivation scores. J Epidemiol Community Health. 1992;46:532–6. doi: 10.1136/jech.46.5.532
Our intention was to emphasize that the knowledge on the importance of social environment on GMP performance is old.
- Piwernetz K. DIABCARE Quality Network in Europe--A model for quality management in chronic diseases. International clinical psychopharmacology. 2001;16 Suppl 3:S5-13. doi: 10.1097/00004850-200104003-00002
The Declaration of Saint Vincent was the starting point for the European DiabCare system. The DiabCare QN was to elaborate a monitoring system. While the project was active, some publications (e.g. 27,29) summarized their recommendations. The dates are determined by the project timeframe.
- Hess BJ, Weng W, Holmboe ES, et al. The association between physicians’ cognitive skills and quality of diabetes care. Acad Med. 2012;87:157–63. doi: 10.1097/ACM.0b013e31823f3a57
It is an old paper but we could not find more recent publication on the physicians’ cognitive skill and DM quality.
Dealing with the indicators for DM monitoring we found very useful these old readings. The references were not changed in the manuscript.
But, as it was suggested to extend the 4.3. Perspectives for Clinical Practice section. To support the new statements, we added 5 new indicators to the references.
Additionally, there was doubled reference for a paper (Piwernetz K. DIABCARE Quality Network in Europe--A model for quality management in chronic diseases. International clinical psychopharmacology. 2001;16 Suppl 3:S5-13. doi: 10.1097/00004850-200104003-00002). Duplication was deleted, and references were renumbered, accordingly.

Round 2
Reviewer 2 Report (Previous Reviewer 2)
Comments and Suggestions for Authors
Dear Authors,
the manuscript is practicaly redy for publication but please add the Strobe Check list in supplementary file and the citation in the references because for scientific community is mandatary.
I suggest in the line 96 this or similar formula "The study adhered to Strobe Statement reporting (Supplemenatry File 1) [21]", and insert in the final statement the indication of the Chck list and the refernces: doi:10.1016/j.jclinepi.2007.11.008.
All other improvment are good.
Author Response
Dear Editor and Reviewer,
Thank you very much – again - for the careful review of our revised manuscript. Please find enclosed the revised version of the manuscript “A composite indicator for primary diabetes care: A cross-sectional study in Hungary” by Undraa Jargalsaikhan, et al.
Your suggestion has been considered. The corresponding changes made in the revised paper. Answers along with the modifications we made are summarized below (comments/questions are in bold capitals).
Sincerely yours, Janos Sandor (on behalf of the authors)
Answer/reflection to the comment of Reviewer-2:
THE MANUSCRIPT IS PRACTICALLY READY FOR PUBLICATION BUT PLEASE ADD THE STROBE CHECK LIST IN SUPPLEMENTARY FILE AND THE CITATION IN THE REFERENCES BECAUSE FOR SCIENTIFIC COMMUNITY IS MANDATARY.
Thank you for this practical advice! The suggested sentence has been inserted into the suggested position. The citation for the suggested publication has been inserted into the reference list. References have been renumbered as needed.
Original text:
“Data regarding GMP structural parameters were extracted from the NHIF’s integrated information system. Socioeconomic status (SES) data were sourced from the 2011 Hungarian Census, which was provided by the Hungarian Central Statistical Office (HCSO).
2.2. Explanatory Variables”
Modified text:
“Data regarding GMP structural parameters were extracted from the NHIF’s integrated information system. Socioeconomic status (SES) data were sourced from the 2011 Hungarian Census, which was provided by the Hungarian Central Statistical Office (HCSO).
The study adhered to Strobe Statement reporting cross-sectional studies [21].
2.2. Explanatory Variables”

This manuscript is a resubmission of an earlier submission. The following is a list of the peer review reports and author responses from that submission.
Round 1
Reviewer 1 Report
Comments and Suggestions for Authors
This study tried to find A composite indicator for primary diabetes care using a cross-sec- 2 tional study in Hungary.
This is a great idea to find only a single indicator for primary diabetes care. This indicator will help authorities to find a way to process the primary DM care performance evaluation, which starts with the evaluation of 31 the composite and adjusted composite measures to identify the GMPs requiring interventions, fol- 32
lowed by experts’ evaluation utilizing detailed information to specify the mode of the required in- 33 tervention.
This study demonstrates for the first time a simple way to find the best solution in a medical practice. But – in the same time – to understand where is the key is necessary to understand this tool.
The methodology is very complex, adequate and well expressed.
In the same time for a new reader will be very difficult to understand, I suggest that 1-2 examples to apply this indicator with a result compared with best practice will help a reader of your article.
Generally speaking, the authors could improve the conclusion of article.
1-2 examples of applied indicator will help this work to be a readable article.
References are appropriate. Most of them are form Diabetes field and / of primary care medicine. The references are carefully chosen, from impact journals. They are updated, from the last 10-12 years, in the area of diabetes. They constitute a real support for the text of the article.
Figures and tables are very easy to understand, support the theory of them indicator. They include mean, standard deviation, statistical significance, logistic regression. The theory is very simple, detailed, complex at the same time, supporting the conclusions. In the same time the maps is a suggestion of impact of this indicator.
But some aspects that are detailed in discussion chapter suggest that this indicator could be improved.
Reviewer 2 Report
Comments and Suggestions for Authors
Dear mìAuthors,
the comments in the annex file.
Best
